# Targeting Interactions between Fibroblasts and Macrophages to Treat Cardiac Fibrosis

**DOI:** 10.3390/cells13090764

**Published:** 2024-04-30

**Authors:** Bo Yang, Yan Qiao, Dong Yan, Qinghang Meng

**Affiliations:** 1Center for Organoid and Regeneration Medicine, Greater Bay Area Institute of Precision Medicine (Guangzhou), School of Life Sciences, Fudan University, Guangzhou 511466, China; yangb22@m.fudan.edu.cn; 2State Key Laboratory of Reproductive Regulation and Breeding of Grassland Livestock, School of Life Sciences, Inner Mongolia University, Hohhot 010021, China; qy@ipm-gba.org.cn; 3State Key Laboratory of Genetic Engineering, School of Life Sciences, Fudan University, Shanghai 200433, China; yandong@fudan.edu.cn

**Keywords:** cardiac fibrosis, myocardial infarction, macrophage, fibroblast, monocytes

## Abstract

Excessive extracellular matrix (ECM) deposition is a defining feature of cardiac fibrosis. Most notably, it is characterized by a significant change in the concentration and volume fraction of collagen I, a disproportionate deposition of collagen subtypes, and a disturbed ECM network arrangement, which directly affect the systolic and diastolic functions of the heart. Immune cells that reside within or infiltrate the myocardium, including macrophages, play important roles in fibroblast activation and consequent ECM remodeling. Through both direct and indirect connections to fibroblasts, monocyte-derived macrophages and resident cardiac macrophages play complex, bidirectional, regulatory roles in cardiac fibrosis. In this review, we discuss emerging interactions between fibroblasts and macrophages in physiology and pathologic conditions, providing insights for future research aimed at targeting macrophages to combat cardiac fibrosis.

## 1. Introduction

In the healthy developing and adult heart, fibrous proteins form the “skeleton” of the heart, known collectively as the extracellular matrix (ECM), provide the myocardial stiffness and mechanical stability required for normal function [1,2]. Fibrosis, however, describes the excessive accumulation of ECM proteins in parenchymal tissue, often reflecting unrestricted activation of repair processes and changes in normal ECM turnover. Myocardial fibrosis, the expansion of the cardiac interstitium and/or perivascular space due to the net accumulation of ECM proteins, accompanies most cardiac pathological conditions [2,3]. Many stimuli can initiate cardiac fibrosis, including myocardial injury, inflammation, and pressure or volume overload. Pathophysiologic changes such as these can lead to increased cardiac stiffness, decreased compliance, and decreased contractility of the heart, resulting in cardiac decompensation and eventually the development of systolic and/or diastolic heart failure [4].

One particular cell type, the cardiac fibroblast, has emerged as the main producer of cardiac ECM proteins during both physiologic and pathologic changes in myocardial structure [5]. Based on mechanical and chemical cues, fibroblasts deposit and remodel the ECM to support tissue function; however, sustained and/or aberrant activation leads to decreased heart function and failure. Immune cell infiltration and activation also play a pivotal role in the development of cardiac fibrosis, and changes in both their number and function can influence its progression. Immune cells secrete factors (including cytokines, growth factors, and matricellular ECM proteins) that directly or indirectly regulate the differentiation of cardiac fibroblasts (CFs) [6]. Macrophages are the major immune cell population found in the resting heart and are located throughout the myocardial interstitium and around blood vessels [7]. As such, targeting immune cells, especially macrophages, may be a beneficial therapeutic avenue to slow or stop the progression of fibrosis in cardiac diseases. This review summarizes the known functions of macrophages in the heart related to fibroblast biology and highlights their potential as therapeutic targets for combating cardiac fibrosis.

## 2. Cellular Composition of the Heart

Single-cell and single-nucleus transcriptome analyses have shown that the adult heart contains at least 11 major heterogeneous populations of cells, including atrial cardiomyocytes, ventricular cardiomyocytes, fibroblasts, endothelial cells, pericytes, smooth muscle cells, immune cells (myeloid and lymphoid), adipocytes, mesothelial cells, and neuronal cells (Figure 1) [8]. Importantly, macrophages account for the majority of immune cells found in the adult heart [7,9,10,11]. Depending on which heart region is being investigated, fibroblasts make up approximately 15–25%, and immune cells 5–10%, of the total population of cells in the heart [8,12]. These observed ratios allow for direct and indirect interactions between fibroblasts and immune cells in the heart, both for homeostatic functions and injury responses [8].

### 2.1. Fibroblasts in Cardiac Tissue

CFs are mesenchymal cells that are derived from the embryonic mesoderm that undergo epithelial-to-mesenchymal (EMT) and endothelial-to-mesenchymal (EndMT) transitions to migrate into and populate the heart early in development (Figure 2) [13,14]. A resident population of fibroblasts is maintained through adulthood, but details of their turnover and life cycle remain unknown. Several studies have suggested that other cell types, including hematopoietic fibroblast progenitors [15,16,17,18], endothelial cells [19,20], and macrophages [21,22], may directly transdifferentiate into fibroblasts or myofibroblasts after injury. However, the limitations of available techniques and models have challenged these theories, and definitive evidence of transdifferentiated cell types contributing to the cardiac fibroblast cell pool is lacking. Interestingly, new genetic tracing models allow for the visualization of multiple lineage-traced fibroblast populations within the same mouse. These observations also substantiate single-cell transcriptome analyses that have identified multiple fibroblast or fibroblast-like populations in the heart. While these are critical breakthroughs in our knowledge of cardiac biology, they uncover complications associated with accurately targeting a desired population of fibroblasts to abrogate fibrosis.

Through cell–ECM and cell–cell interactions, fibroblasts respond to changes in mechanical tension and chemical cues from their surrounding environment. Due to their regulation of cardiac ECM proteins, fibroblasts are thought to be central to age-related changes in myocardial structure and content, including collagens, proteoglycans, fibronectin, laminins, elastin, and growth factors (Figure 1) [23]. The differentiation of CFs into secretory, matrix-producing, contractile cells, called myofibroblasts, is the main source of ECM protein dysregulation in fibrotic pathogenesis. Under normal physiological conditions, CFs do not exhibit significant proliferative activity or collagen synthesis, and the ECM network is maintained through a dynamic balance of protein synthesis and breakdown. In the pathological state, fibroblasts are stimulated by mechanical forces, cytokines, and chemokines to proliferate and differentiate into myofibroblasts, resulting in altered regulation of ECM proteins such as collagens, fibronectin, and matrix metalloproteinases (MMPs) [24]. Over time, interstitial collagen density increases, the ratio of different collagen types is severely disordered, and ECM alignment is disturbed, all of which are hallmarks of cardiac fibrosis and structural remodeling [25]. Although investigation of fibroblast–myocyte interactions has dominated cardiac fibrosis research, how fibroblasts interact with and are affected by immune cells, specifically macrophages, has emerged as a critical area of research in cardiac biology.

### 2.2. Macrophages in Cardiac Tissue

In recent years, macrophages have become increasingly attractive as potential targets for improving myocardial cell survival and repair, and they have been evaluated in several preclinical studies [26]. Zebrafish hearts can regenerate throughout adulthood, in part because their cardiomyocytes are mononuclear and remain proliferative throughout the life of the fish [27]. However, it has also been shown that cardiac resident macrophages are critical for zebrafish cardiac regeneration due to their ability to support revascularization, CM survival and replenishment, and diminish scarring [28]. Studies in mice indicate that the mammalian heart possesses significant regenerative potential during embryonic and neonatal life, but in contrast to zebrafish, their regenerative capacity is rapidly lost after birth [29]. Interestingly, a cell depletion model in mice reveals that cardiac regeneration and neoangiogenesis after neonatal MI are dependent upon macrophages [20]. Neonates depleted of macrophages were unable to regenerate their myocardium and form fibrotic scars, resulting in reduced cardiac function and angiogenesis [30].

The classification of resident tissue macrophages into pro-inflammatory M1-type and anti-inflammatory M2-type, a paradigm that prevailed until around two decades ago, has become obsolete with advances in single-cell sequencing technologies [31]. It is now understood that macrophages exhibit diverse developmental and tissue-specific functional profiles. Their extensive heterogeneity has a significant regulatory impact on various diseases [32]. Macrophages in the heart are a heterogeneous cell population derived from different developmental lineages (Figure 3), including monocyte- and embryonic-derived subpopulations [33,34]. Monocyte-derived macrophages that originate from hematopoietic progenitors are recruited to tissues through established chemokine signaling pathways [35,36]. In contrast, embryonic-derived macrophages arise from several potential lineages, including primordial yolk sacs, recombinant activation gene 1 (Rag1)^+^ lymphoid bone marrow cells, and fetal liver mononuclear cell progenitors [37,38,39,40]. The earliest macrophages are thought to originate from bipotential red lineage progenitor cells located in the early yolk sac [38]. Yolk sac-derived monocytes migrate outside the yolk sac during early embryonic life and populate developing organs such as the heart, which contains a large number of embryonic-derived macrophages [41,42].

The cell-surface CC chemokine receptor-2 (CCR2) protein is an important identifier for macrophage subsets in the heart. The developing heart contains both CCR2^−^ and CCR2^+^ macrophages derived from a complex array of embryonic macrophages [43]. CCR2^−^ resident cardiac macrophages are derived from primitive yolk sac or fetal monocyte progenitors, are maintained by local proliferation without monocyte input, and are associated with coronary artery growth and neonatal heart regeneration [41,42]. These cells migrate to the heart after developing from the yolk sac, and they specifically express CX3CR1 [44]. In contrast, CCR2^+^ resident cardiac macrophages originate from hematopoietic progenitors and are recruited to the myocardium during the first few weeks of life [41,43]. It has been demonstrated that resident CCR2^−^ macrophages inhibit cardiac fibrosis, whereas recruited monocyte-derived macrophages stimulate cardiac fibrosis, and that the depletion of resident macrophages leads to accelerated development of heart failure [45]. Another study showed that CCR2^+^ resident cardiac macrophages promote the recruitment of pro-inflammatory leukocytes that exacerbate myocardial injury and mediate dysfunction, whereas CCR2^−^ resident cardiac macrophages protect the damaged heart from adverse remodeling [46]. While additional research is needed, the identification of disparate populations of cardiac macrophages with targetable surface proteins is important for developing cell-based therapeutics against cardiac fibrosis.

## 3. Interactions between Cardiac Macrophages and Fibroblasts

Macrophages and fibroblasts in the heart interact through various molecular pathways (Figure 4). Activated fibroblasts appear to maintain *Csf1* expression during fibrosis and inflammation. Whereas the *Csf1* receptor (*Csf1r*) is expressed primarily on macrophages, the persistence of CSF1 provides a critical signal for macrophages of different origins to remain in fibrotic lesions and to regulate the proliferation, differentiation, migration, and activation of target immune cells through the PI3K-AKT, JAK-STAT, and MAPK pathways [47]. Reciprocally, activated macrophages promote CSF1 protein production by fibroblasts [48] and secrete growth factors such as platelet-derived growth factors (PDGFs) and AREG that promote fibroblast proliferation [49,50]. Macrophages effectively drive the fibrogenic program through the expression of transforming growth factor-β (TGF-β) ligands, and they direct fibroblast activation in concert with IL-6, IL-17, and AREG (Figure 4) [49]. On the other hand, macrophages are also essential for the resolution of fibrosis [51]. Notably, activated macrophages can degrade ECM proteins through extracellular proteolysis [52,53], or they secrete milk fat globule EGF factor 8 (MFGE8) to target collagen for internal uptake and degradation in lung tissue [54]. Thus, depending on the nature of injury, inflammation, and repair, fibroblast–macrophage circuits can affect heart function in a variety of ways. In this regard, the disruption of cellular communication that mediates ECM deposition and breakdown may determine the severity and duration of fibrosis and thus disease outcomes. It is crucial to highlight that fibrosis in the heart markedly differs from fibrosis in other organs. Cardiac fibrosis is notably resistant to resolution, unlike fibrosis in tissues with higher regenerative capabilities, such as the liver and skin. Consequently, identifying effective therapeutic targets for cardiac fibrosis is of heightened importance.

Whether CFs influence the process of macrophage polarization and activation in cardiac remodeling after injury is an ongoing focus of research. Isolated CFs pre-treated with TGF-β1 or LPS were co-cultured with monocytes, and their effects on macrophage polarization were evaluated using flow cytometry and cytokine secretion. LPS-treated CFs induced M1-like polarization of monocytes and secretion of pro-inflammatory cytokines, including TNF-α, IL-12, and MCP-1. On the other hand, CFs treated with TGF-β1 stimulated the differentiation of monocytes to M2-like macrophages, which included an increase in IL-10 and a decrease in IL-12 protein secretion. This study is the first to show that CFs can secrete cytokines that recruit monocytes and induce their differentiation [55]. Additional research has aimed to determine the relevance of these signaling networks to fibroblast and macrophage behaviors in vivo, and several proteins have emerged as critical factors in their communication during the onset and progression of cardiac fibrosis. The majority of the interacting molecules highlighted in this review are secreted by either cardiac fibroblasts or macrophages [56,57]. Additionally, there are molecules whose specific cellular origins remain unidentified but still indirectly affect the interplay between these two cell types. 

### 3.1. TGFβ

TGF-β plays an important role in cardiac hypertrophy and cardiac fibrosis by activating fibroblasts and promoting collagen production [58]. TGF-β from macrophages or other sources (including cardiomyocytes and fibroblasts themselves) can act as an accessory signal for macrophage–fibroblast interactions [58,59]. It has been shown that when diastolic dysfunction occurs, cardiac macrophages produce IL-10, which activates fibroblasts to proliferate and deposit collagen, further contributing to impaired myocardial relaxation and increased myocardial stiffness (Table 1) [60]. A recent study also showed that IL-10 and TGF-β are produced by cardiac macrophages in response to stress overload, and that they stimulate myofibroblast differentiation and collagen production (Table 1) [45]. In fibroblasts, TGF-β induces the expression of many factors indicative of activated fibroblasts or myofibroblasts, including smooth muscle α actin (αSMA), collagen I, cartilage oligomeric matrix protein (Comp), periostin, and connective tissue growth factor (CTGF), largely via SMAD3 activation downstream of TGF-β receptor 1/ALK5 activation [61]. Furthermore, macrophages stimulated with IL-13 and the proinflammatory cytokine TNFα expressed significantly more TGF-β1 compared to macrophages stimulated with IL-4, IL-13, or IL-4 combined with TNF-α [62]. The IL-13/TGF-β1 signaling axis has been recognized as a key pathway for fibrosis in inflammatory immune diseases. In studies of mouse heart transplantation, IL-13-induced TGF-β1 production through IL-13Rα_2_ signaling, which resulted in significantly more collagen deposition and led to fibrosis in allogeneic cardiac grafts, was prevented by treatment with siRNA for IL-13Rα_2_ [63]. These data all demonstrate the important regulatory role of TGF-β between fibroblasts and macrophages during cardiac fibrosis. To facilitate a concise overview, we have summarized and organized the comprehensive array of molecular mechanisms governing the interactions between cardiac fibroblasts and macrophages in Table 2.

### 3.2. IL-4 and IL-6

In vivo studies have demonstrated that administration of IL-4 in mice post-myocardial infarction (MI) selectively augments the population of M2-like macrophages. This elevation correlates with an enhanced activation of fibroblasts, which contributes to an improved prognosis for MI. Notably, this treatment does not alter the extent of fibrosis within the border zone or the distal regions of the infarct [65,70]. Interestingly, in co-culture experiments with macrophages and fibroblasts, M1 macrophages were found to promote fibroblast activation, whereas M2 macrophages promoted fibroblast proliferation [70]. This indicates that the reparative response that is commonly attributed to the M2 phenotype might initially be instigated by the M1 phenotype, with fibroblasts serving as the critical intermediary in this biphasic process. Thus, inflammation, through the activation of fibroblasts, may precede fibrosis, with a subsequent transient proliferation of fibroblasts occurring during the repair phase [70]. M2-like anti-inflammatory macrophages protect cardiomyocytes, promote neovascularization, and mediate cardiac repair after MI by reducing the overall inflammatory response [65,77]. RNA sequencing studies have determined the existence of a spectrum of macrophage populations defined within the M1/M2 activation classifications, suggesting that macrophages are flexible enough to respond differently to different environments [78,79,80]. It has also been shown that myeloid cells are a key target cell type for IL-4Rα signaling during cardiac remodeling after MI. Bone marrow-specific knockdown of IL-4Rα in mice resulted in reduced collagen 1 deposition, upregulation of MMPs, and downregulation of tissue inhibitors of metalloproteinases (TIMPs), leading to dysregulation of inflammation and insufficient fibrotic remodeling, which ultimately worsened cardiac function after MI [81].

IL-6 is a key mediator in cardiac fibrosis [82,83], although its precise involvement remains unknown. It plays an important role in cardiomyocyte hypertrophy and myocardial fibrosis as a downstream signal of hypoxia-induced mitogenic factor (HIMF) through activation of MAPK and CaMKII (Ca^2+^/calmodulin-dependent protein kinase II)-STAT3 pathways [71]. In addition, the IL6/STAT3 pathway is involved in regulating pressure overload-induced heart failure in a mouse model [84]. Interestingly, IL-6 was expressed at low mRNA levels in cardiac fibroblasts and not in cardiomyocytes or macrophages. Macrophages are required for IL-6 production by fibroblasts in vitro, although the mechanism is unknown [85]. A co-culture of macrophages with cardiac fibroblasts stimulated TGF-β1 activation and phosphorylation of Smad3, and IL-6-neutralizing antibodies blocked this event [85]. Administration of angiotensin II (Ang II) to mice was shown to upregulate IL-6 expression in the heart [85]. Interestingly, *Il-6^−/−^* mice show reduced expression of α-SMA, TGF-β1, and collagen I in the heart, which reduces Ang II-induced cardiac fibrosis [85]. While their signal transduction pathways may be well-studied overall, exactly how IL-4 and IL-6 may contribute to fibroblast–macrophage interactions in the heart requires further investigation.

### 3.3. IL-17A

IL-17A may play an important regulatory role in cardiac fibroblast–macrophage interactions. Based on the expression of the cell surface marker lymphocyte antigen 6C (Ly6C), mouse monocytes were divided into three subpopulations: Ly6C^high^, Ly6C^middle^, and Ly6C^low^ [86,87]. The Ly6C^high^ and Ly6C^middle^ monocyte subsets have pro-inflammatory functions [88,89], while Ly6C^low^ monocytes have patrolling and anti-inflammatory functions [88,90]. IL-17A induces the production of chemokines by CFs, leading to an infiltration of neutrophils and Ly6C^high^ monocytes/macrophages (MO/MΦs) in the heart. IL-17A directs the conversion of Ly6C^high^ MO/MΦ to a more pro-inflammatory phenotype via CF-derived GM-CSF [64]. Other regulatory fibrosis pathways were upregulated in Ly6C^high^ MO/MΦs, such as thrombospondin-1 (*Thbs1*), which activates latent TGF-β bound in the ECM and initiates TGF-β-dependent signaling pathways [91]. In contrast, during acute injury in autoimmune myocarditis, activated fibroblasts not only inhibit the differentiation of monocytes from Ly6C^low^ into reparative macrophages through the upregulation of IL-17A but also promote the shedding of MER receptor tyrosine kinase (MerTK) on Ly6C^high^ from macrophages, leading to an overall proinflammatory phenotype with impaired efferent cell activity [92].

### 3.4. MMP-2, MMP-9, and MMP-12

MMPs are a highly conserved family of proteolytic enzymes that have specificity for particular ECM proteins, including collagen and elastin [93]. During reparative cardiac healing, specific MMPs activate and degrade the preexisting ECM to disrupt the fibrillar collagen network, allowing inflammatory cells such as neutrophils and macrophages to infiltrate into the infarcted tissue and remove necrotic myocytes [72]. Earlier studies have identified an increase in MMP-2 and MMP-9 in myocardial tissue in a time-dependent manner after acute MI in mice, and their overexpression (mainly by neutrophils and macrophages) may lead to excessive ECM degradation during the early phase of MI. This impairs infarct healing and exacerbates early remodeling, which can lead to cardiac rupture [72].

In recent years, it has been found that three days after MI in mice, millions of monocytes that express Ly6C^high^ migrate to the infarcted heart to drive the inflammatory response. These cells then differentiate into macrophages with high phagocytic and protein hydrolytic activities and produce IL-1, IL-6, and TNF [94,95,96,97]. Approximately 4 days after MI, reparative Ly6C^low^ macrophages begin to accumulate at the site of injury and secrete vascular endothelial growth factor (VEGF), TGF-β, and IL-10 to stimulate angiogenesis and myocardial repair processes [94,98,99]. One mechanism for this phenomenon is that M2 macrophages promote wound healing by secreting MMP-12 and inhibiting neutrophil migration [94]. *MMP-12^−/−^* mice show increased neutrophil numbers, upregulated MMP-9, reduced fibrosis and myofibroblast numbers, and impaired cardiac function and survival (Table 1) [69]. Importantly, resident cardiac macrophages but not monocyte-derived macrophages were responsible for improved wound healing and cardiac function after MI [9]. Thus, despite the similar phenotype of monocyte-derived macrophages, the reduction in resident macrophages after MI leads to poor remodeling and deterioration of cardiac function [9].

### 3.5. CX3CR 1

Fractalkine (CX3CL1) is the only member of the CX3C chemokine subfamily, and its G-protein-coupled receptor, CX3CR1, is present in inflammatory cells as well as cardiomyocytes [100]. Phagocytic macrophages with increased CX3CR1 expression can alter the activity of CFs, leading to reduced ECM content in the marginal zone and enhanced cardiac contractility [73]. Three days after injecting bone marrow-derived monocytes into the heart following ischemia/reperfusion (I/R), macrophage subtypes shifted from mainly the CCR2^−^CX3CR1^+^ phenotype observed in basal conditions to a mixed population of CCR2^+^ and CCR2^+^CX3CR1^+^ macrophages, suggesting that the overall macrophage profile in the heart shifted to repair. In stem cell therapy experiments in mice after ischemia–reperfusion, CCR2^+^ or CX3CR1^+^ macrophages were isolated from the heart 7 days after I/R and cultured with freshly isolated cardiac fibroblasts for 72 h. A gene expression analysis showed that CCR2^+^ macrophages increased smooth muscle α-actin (Acta2/αSMA), collagen 1α2 (Col1a2), and lysyl oxidase (Lox) in fibroblasts. In contrast, co-culture with CX3CR1^+^ macrophages slightly decreased the expression of these genes but increased the connective tissue growth factor (Ctgf). This suggests that specific macrophage subtypes mobilized by stem cell therapy influence the activity of cardiac fibroblasts and, consequently, the passive mechanical properties of the infarct zone of the heart [65,101].

### 3.6. microRNA-21

MicroRNAs are small molecular RNAs that do not code for proteins but do play a role in the pathogenesis of many cardiovascular diseases [102]. As a component of exosomes, they may mediate intercellular communication in a paracrine form [103]. Among them, miR-21 is one of the most strongly expressed miRNAs in various cardiac cell types, and it is preferentially expressed in non-cardiomyocyte cells and upregulated in various cardiac diseases associated with cardiac fibrosis [104,105]. It has been shown that microRNA-21 inhibits ERK signaling and enhances cardiac fibroblast survival by suppressing sprouty homolog 1 (SPRY1) expression in CFs, which has implications for the overall structure and function of the heart [74]. Targeted gene deletion of microRNA-21 in mouse macrophages prevented their proinflammatory polarization and reduced subsequent pressure overload-induced cardiac fibrosis and dysfunction [106]. This study shows that macrophage-derived microRNA-21 may control myocardial fibrosis through intercellular communication with fibroblasts. Furthermore, a systematic assessment of ligand–receptor-mediated intercellular communication across all cell types has revealed that miR-21 is a predominant regulator of macrophage–fibroblast signaling, and it facilitates the transformation of quiescent fibroblasts into active myofibroblasts. Similarly, macrophages cultured in vitro have demonstrated equivalent capabilities in modulating fibroblast activation via miR-21 [106].

### 3.7. microRNA-155

MicroRNA-155 has also been found to play a regulatory function in macrophage–fibroblast interactions. In a mouse model of MI, macrophages and fibroblasts showed increased expression of microRNA-155, whereas the primary transcript pre-microRNA-155 was expressed only in macrophages [75]. MicroRNA-155 was found to inhibit cardiac fibroblast proliferation by downregulating son of sevenless 1 (*Sos1*) expression, and it was shown to promote inflammation by decreasing cytokine signaling inhibitor 1 expression [75]. In vivo, microRNA-155 knockout mice exhibit increased fibroblast proliferation and collagen production, as well as reduced inflammation in damaged hearts [75]. MicroRNA-155 inhibitors may modulate acute MI and related adverse events.

### 3.8. TLR2

Toll-like receptors (TLRs) are pattern-recognition receptors of the innate immune system. TLR2 is expressed only on myeloid-derived cells, including monocyte-derived macrophages [107]. TLR2 has been shown to play a key role in cardiovascular disease, and knockdown or inhibition of TLR2 with neutralizing antibodies in mice attenuated Ang II-induced myocardial fibrosis [76]. The authors showed that TLR2 deficiency inhibits macrophage-dependent CF activation via modulation of the TGF-β/Smad2/3 pathway [76]. In addition, another study confirmed that TLR2 knockdown through MAPKs/NF-κB signaling significantly reduced isoproterenol (ISO)-induced cellular inflammation and cardiac remodeling. ISO significantly increased TLR2-myeloid differentiation factor 88 (MyD88) signaling in macrophages and cardiomyocytes in a TLR1-dependent manner. Furthermore, DAMPs, such as HSP70 and fibronectin 1 (FN1), were released from cells upon ISO stimulation, which further activated TLR1/2-MyD88 signaling and subsequent pro-inflammatory cytokine expression and cardiac remodeling. The above studies suggest that TLR2 can mediate fibroblast function and the progression of cardiac fibrosis through the regulation of macrophage activity and the inflammatory response. Temporal TLR2 inhibition may be a viable avenue to control the progression of hypertensive heart disease and other inflammatory diseases.

## 4. Fibroblast–Macrophage Interactions for Novel Fibrosis Treatments

Myocardial fibrosis is pervasive in heart diseases of many pathologic origins and is associated with the occurrence of adverse cardiovascular events and worsened patient outcomes. However, there is no known treatment for alleviating fibrosis in the heart [108]. The drugs currently used in the treatment of heart failure, including angiotensin receptor-neprilysin inhibitors (ARNI) [109,110], sodium–glucose cotransporter-2 inhibitors (SGLT2i) [111,112], and spironolactone [113], have been shown to be only mildly effective in alleviating cardiac fibrosis and improving ventricular remodeling. Other drugs used in heart failure therapy include torasemide [114], soluble guanylyl cyclase (sGC) agonists [115], and pirfenidone [116], which is already marketed as a non-heart failure therapy, lack large-scale clinical studies to validate their efficacy and safety in the treatment of cardiac fibrosis. Notable among them are therapies that inhibit the renin–angiotensin–aldosterone system (RAAS), such as candesartan, spironolactone, eplerenone, and losartan [117,118,119,120,121]. Clinical trial data suggest that RAAS inhibitors can reduce the concentration of the amino-terminal peptide of type III procollagen and the carboxy-terminal peptide of type I procollagen circulating in patients with cardiac disease, implying more balanced ECM deposition. However, these treatments have been shown to only moderately abrogate fibrosis, not to reverse it in patients with established fibrosis and/or cardiomyopathy [122]. Pirfenidone was the first drug to demonstrate efficacy in idiopathic pulmonary fibrosis (IPF) in a replicated, randomized, placebo-controlled phase III clinical trial [123,124]. It is being tested in clinical studies for a variety of diseases, including heart failure with a preserved ejection fraction [116,125,126]. It is a potent cytokine inhibitor, inhibiting collagen synthesis by decreasing the expression of the pro-fibrotic factor TGF-β and the pro-inflammatory cytokines TNF-α and interleukin [127,128]. However, its pronounced adverse effects have limited its clinical application thus far, and further trials are needed to determine whether safe and efficacious treatment is possible [129]. Several possible targets for the treatment of cardiac fibrosis have emerged in recent years, including the TGF-β signaling pathway [130,131], fibronectin [59], HMGA1 [132], and BET [133]. The continued development of safe and effective new drugs for the treatment of cardiac fibrosis remains an important goal of cardiovascular disease research.

Clinical trials of stem cell-based therapies are ongoing, and it is not clear which ingredient is responsible for the therapeutic effects without significant efficacy. There are a number of issues that need to be addressed in this process, such as low cardiac homing of stem cells, weak paracrine secretion, and low survival rates [134,135,136,137]. Therefore, direct reprogramming of fibroblasts has become a possible alternative therapy, which involves the in situ trans-differentiation of fibroblasts into cardiomyocytes to repopulate the cardiac scar to restore cardiac function, a process that is both difficult and time-consuming [138,139]. Three cardiac-derived transcription factors, Gata4, Mef2c, and Tbx5 (GMT), can induce direct reprogramming of mouse fibroblasts into induced cardiomyocytes (iCMs). Additional factors such as Mesp1 and Myocd are required in humans [140]. One of the notable problems faced is that inflammation and the immune response can hinder the reprogramming process in mice [140]. In future studies, if immune cells can be targeted for intervention, it may be possible to greatly improve reprogramming efficiency. Direct cardiac reprogramming needs to be improved if it is to be used in humans, and the molecular mechanisms involved remain difficult to understand. Further advances in cardiac reprogramming research will bring us closer to cardiac regenerative therapies.

In another significant research avenue, efforts to utilize biomaterials for cardiac fibrosis treatment date back to at least 20 years ago [141]. Acknowledging the challenge of retaining and ensuring the survival of transplanted living cells post-MI in ischemic tissues, one approach involved using fibrin glue to co-inject skeletal myoblasts into the ischemia-reperfused heart tissue of rats. At a five-week follow-up, the areas populated by skeletal myoblasts were substantially larger, while the infarct size and myoblast size were significantly reduced in the rats treated with fibrin glue compared to the controls [141]. As research has evolved, animal studies have indicated that even the sole injection of fibrin glue biomaterial into rat cardiac muscle tissue can ameliorate cardiac function post-MI [142]. In more recent years, decellularized extracellular matrix (ECM) materials fashioned into hydrogels have demonstrated considerable potential, showing beneficial therapeutic effects in animal models of myocardial injury and confirming safety in phase 1 clinical trials. It has been observed that myocardial matrix hydrogels can effectively remove reactive oxygen species (ROS) from damaged myocardial tissues, lessen mitochondrial superoxide content, and increase the incorporation of thymidine analogs, thus creating a conducive microenvironment for DNA synthesis in cardiomyocytes. ROS are known to be significant in the proliferative activation of cardiac fibroblasts, prompting them to express more hypoxia-inducible factor 1α (HIF-1α) [143]. ROS can also activate matrix metalloproteinases (MMPs), leading to ECM remodeling, which can result in detrimental outcomes [144]. Hence, myocardial matrix hydrogels aim to support DNA repair and cell cycle activation while mitigating ROS-induced cardiac tissue damage, thereby improving cardiac injury outcomes [145]. Despite these advancements, there remain challenges to the standardized clinical application of biomaterials. Toxicological assessments of various biomaterials are still not comprehensive, and the duration of animal testing is relatively short, necessitating further time to verify their long-term safety and efficacy.

The recent development of chimeric antigen receptor T-cell immunotherapy (CAR-T) has propelled the efficacy and application of cell-based therapeutics. CAR-T therapy consists of modifying a patient’s T cells with lentiviral or retroviral gene transduction and growing a large number of these cells in culture. When introduced back into the patient, the engineered T cells can bind to a specific antigen on the surface of target cells, like tumors, to induce apoptosis [146]. Driven by breakthroughs in cancer therapy, scientists have sought to determine the feasibility of targeting activated CFs with this technology. Excitingly, recent work has demonstrated the effectiveness of CAR-T therapy to specifically target endogenous activated CFs expressing the cell surface molecule fibroblast activating protein (FAP) [147]. Furthermore, advancements in lipid nanoparticle (LNP) packaging have allowed for efficient T-cell transduction in vivo, as researchers have recently encoded an anti-FAP CAR by packaging FAP mRNA into CD5-targeted LNPs [148]. Treatment with modified mRNA-targeted LNPs accurately targeted CFs to induce cell death and significantly reduced fibrosis and restored cardiac function after injury in mice. Importantly, CAR-T cells did not persist in the heart or other tissues long-term. However, it remains unknown whether FAP-expressing fibroblasts are a suitable cell population to target at all stages of fibrotic progression or if a narrow therapeutic window exists. Temporal control of cell activation and behavior is essential for decreasing off-target and adverse side effects, and CAR-T therapies may be flexible enough to provide a solution. As this review has outlined, interactions between fibroblasts and macrophages may serve to target fibrosis-related cell activation more specifically. As such, an attractive approach for CAR-T therapies would be to use unique macrophage surface markers to target macrophages and indirectly control cardiac fibroblast activation, breaking the fibrotic cycle of ECM disruption and inflammation.

Recent advancements in protein-based cell characterization techniques have uncovered complex interactions between macrophages and fibroblasts that can be tracked through analyses of ligand–receptor expression networks [149]. Receptor-mediated interactions between macrophages and fibroblasts (e.g., TGF-β from macrophages with TGF-βR expression on fibroblasts) may allow for therapies that can more specifically target the regulation of cellular activation and ultimately the progression of cardiac fibrosis. Advanced proteomic analyses such as these also provide a valuable reference for specific cell surface molecules found on macrophages and CFs that can be utilized for targeted therapeutics, such as CAR-T therapy, discussed above. These analyses can help turn difficulties associated with macrophage and CF heterogeneity into an advantage for targeting precise aspects of disease progression and injury responses.

Another technological advancement is the development of functional organoid model systems derived from human pluripotent stem cells. These in vitro models help compensate for many limitations that exist in terms of genetic and physiological differences between animal models and humans. Notably, a post-MI organoid model was recently created [150]. This model can simulate the myocardial response to infarction, where multiple cell types in three-dimensional space are essential to cellular responses after injury, including fibrosis and pathological calcium handling. One major limitation of cardiac organoid models, however, is that they lack inflammatory cells [150], which is a critical component of heart function and failure. Similarly, hPSC-derived self-organized “cardioids” can reproduce heart chamber-like morphogenesis in the absence of non-cardiac tissue [151] and serve to study mechanisms of human cardiogenesis and heart disease. However, this model uses only hPSCs from the first heart field lineage and the earliest stages of cardiogenesis and therefore does not yet faithfully reproduce most cardiac defects. Overall, the future possibility of co-culturing myocardial infarct-like organs with immune cells to mimic the microenvironment in vivo and thus clarify the pathways of immune cell action in cardiac fibrosis remains a challenge to be addressed.

## 5. Conclusions

In summary, macrophage–fibroblast interactions are critical to heart form and function. The discovery of therapies that target specific ways in which macrophages and fibroblasts communicate may provide viable approaches for mitigating fibrosis and the progression to heart failure.

## Figures and Tables

**Figure 1 cells-13-00764-f001:**
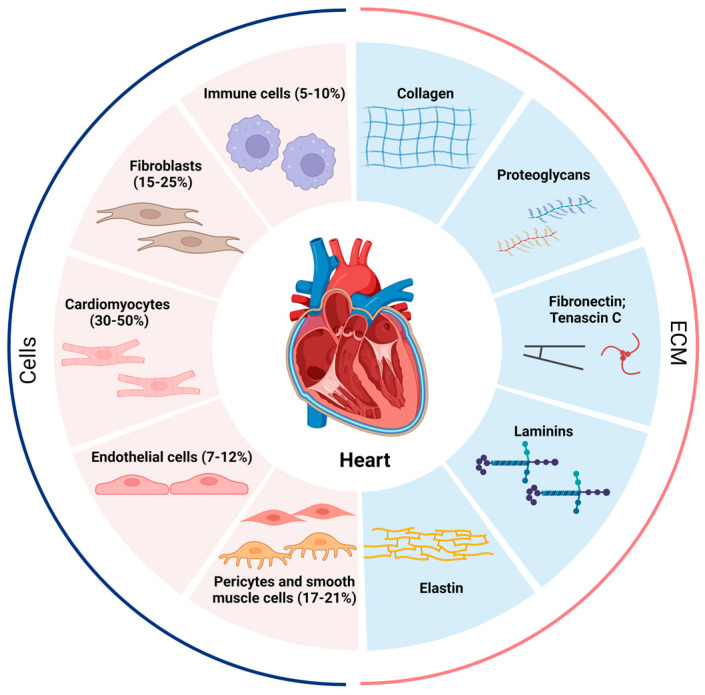
Proportion of various cells in the heart and composition of the ECM. The human heart at homeostasis consists of roughly 5–10% immune cells, 15–25% fibroblasts, 30–50% cardiomyocytes, 7–12% endothelial cells, and 17–21% pericytes and smooth muscle cells, and the cellular proportions change dramatically when the heart is under stress or other stimuli. The extracellular matrix (ECM) is secreted mainly by fibroblasts and contains collagen, proteoglycans, fibronectin, tenascin C, laminins, and elastin. The pathological increase in fibroblasts and the abnormal deposition of ECM lead to the development of cardiac fibrosis.

**Figure 2 cells-13-00764-f002:**
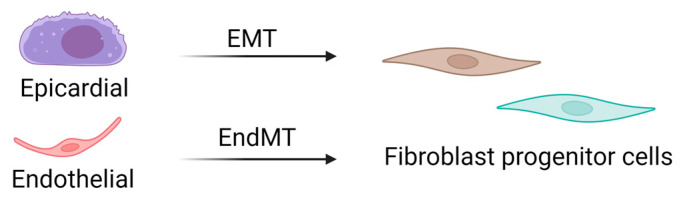
Source of cardiac fibroblasts. Cardiac fibroblasts are derived from mesenchymal progenitor cells, most of which are derived from epithelial–mesenchymal transitions. In contrast, fibroblasts in the septum and right ventricle are derived from endothelial–mesenchymal transitions. EMT: epithelial-to-mesenchymal transition; EndMT: endothelial-to-mesenchymal transition.

**Figure 3 cells-13-00764-f003:**
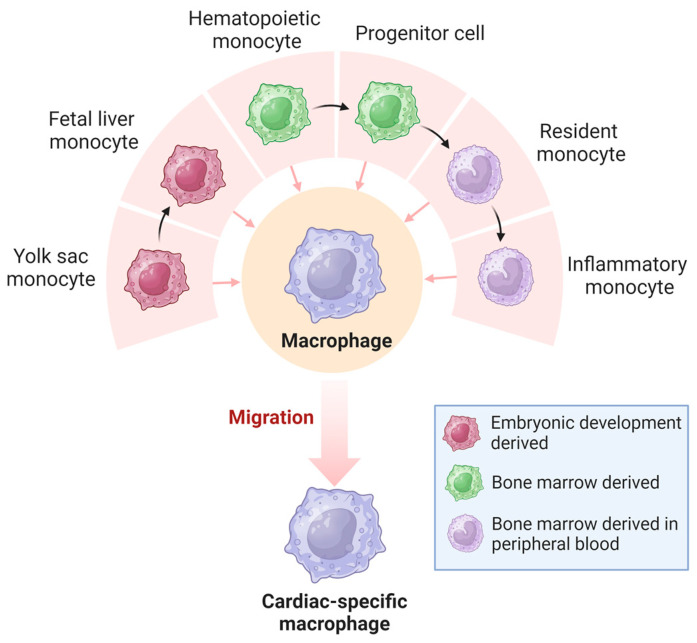
Origin of cardiac macrophages. Macrophages in the heart originate from the yolk sac and bone marrow and become highly heterogeneous cardiac-specific macrophages.

**Figure 4 cells-13-00764-f004:**
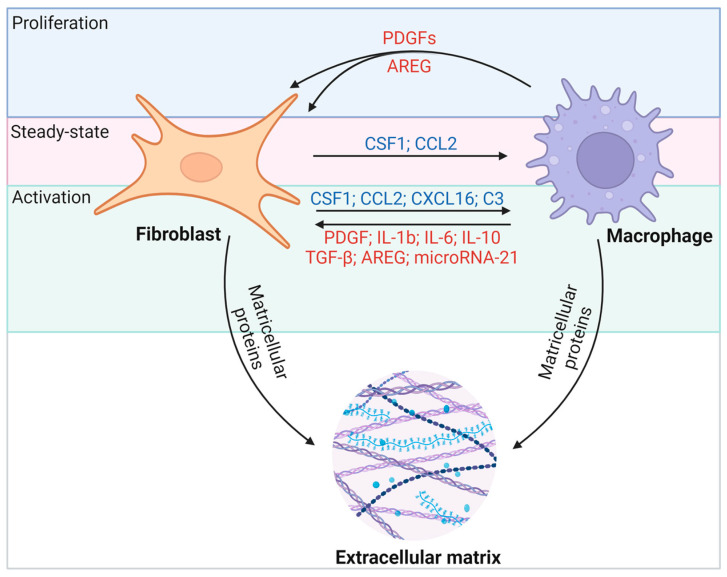
Macrophages and fibroblasts interact through multiple molecular pathways. Fibroblasts recruit macrophages to fibrotic foci through the secretion of cytokines such as CSF1 and the chemokine CCL2, which leads to proliferation, activation, and differentiation. CSF1–CSF1R interactions are thought to be a persistent force linking fibroblasts and macrophages under different conditions. In turn, activated macrophages secrete TGF-β, IL-6, and AREG to stimulate fibroblast proliferation and activation. Among them, the IL-13/TGF-β1 signaling axis has been identified as a key pathway for fibrosis in inflammatory immune diseases.

**Table 1 cells-13-00764-t001:** Studies of fibroblast–macrophage interactions in cardiac fibrosis.

Cardiovascular Disease	Model	Molecular Mechanisms	Main Outcome	References
**TAC-induced cardiac pressure overload**	C57BL/6j and *CCR2^−/−^* mice	Increased release of TGF-β1 and IL-10 from cardiac macrophages in TAC mice	Myofibroblast differentiation and collagen production	[45]
**Cardiac fibrosis with diastolic dysfunction**	Patients with hypertension and HFpEF; 8–30 wk-old *CCR2^−/−^, CX3CR1^CreER^* and *IL10^fl/fl^* C57BL/6 mice	IL-10 contributes to a macrophage phenotype shift toward a profibrotic subset, which activates fibroblasts	Cardiac fibrosis with diastolic dysfunction	[60]
**Inflammatory dilated cardiomyopathy (DCMi)**	6–10 wk-old WT, *IL17ra^−/−^* BALB/cJ and CBy.PL(B6)-Thy1a/ScrJ (Thy1.1) founder mice	IL-17A induces the production of GM-CSF by CFs, leading to infiltration of Ly6C^high^ MO/MΦs	IL-17A directs the conversion of Ly6C^high^ MO/MΦ trans to a more pro-inflammatory phenotype via CF-derived GM-CSF	[64]
**MI**	6–8 wk-old C57BL/6 mice and *Trib1^–/–^* mice of a mixed background of C57BL/6 and SV129	IL-4 treatment increased the number of cardiac M2-like macrophages, which increased the activation of CFs	IL-4 is a potential biological drug for treating MI	[65]
**MI**	7–10 wk-old female WKY rats	CDCs reduce the number of CD68^+^ macrophages within the ischemic heart	CDC limits acute injury and attenuates cardiac fibrosis	[66]
**MI**	6–8 wk-old female WKY rats	Reduced levels of IL-1β and TNF-α in the peri-infarct region	CSps enhance cardiomyocyte proliferation and angiogenesis and attenuate hypertrophy and fibrosis	[67]
**AMI**	Yucatan mini-pigs	/	IC delivery of allo-CDCs is safe, feasible, and effective in cardioprotection, reducing IS, preventing MVO, and attenuating adverse acute remodeling	[68]
**MI**	8–10 wk-old WT male C57BL/6 and *MMP12^−/−^* mice	Significantly increased mRNA expression of CXCL1, CXCL2, and CXCL5 in *MMP12^−/−^* mice	MMP-12 production by Ly6C^low^ macrophages promotes wound healing	[69]

TAC, transverse aortic constriction; MI, myocardial infarction; CDCs, cardiosphere-derived cells; CSps, cardiospheres; WKY, Wistar–Kyoto; AMI, acute myocardial infarction; IC, intracoronary; IS, infarct size; MVO, microvascular obstruction; WT, wild-type.

**Table 2 cells-13-00764-t002:** Molecular mechanisms of interaction between cardiac fibroblasts and cardiac macrophages.

Molecules	Cellular Origin	Molecular Mechanisms	References
**TGF-β**	Macrophages, cardiomyocytes, and fibroblasts themselves	Activation of downstream SMAD3 via TGF-β receptor 1/ALK5 in fibroblasts induced activated fibroblasts to express αSMA, collagen I, Comp, periosteal proliferator protein, and CTGF	[61]
**IL-4**	/	Inhibition of the increase in the number of M2-like macrophages and increase in the activation level of fibroblasts improve the prognosis of MI	[65,70]
**IL-6**	Macrophages and fibroblasts	IL-6 acts as a downstream signal for HIMF and activates the MAPK and CaMKII-STAT3 pathways	[71]
**IL-17A**	/	IL-17A induces the production of chemokines by CFs, leading to an infiltration of neutrophils and Ly6Chigh MO/MΦs in the heart	[64]
**MMP-2, MMP-9, MMP-12**	Neutrophils and macrophages	MMP-2 and MMP-9 over-grade the ECM in the early stages of MI. MMP-12^−/−^ mice show increased neutrophil numbers, upregulated MMP-9, and reduced fibrosis and myofibroblast numbers	[69,72]
**CX3CR 1**	Macrophages	Altered activity of CFs, resulting in decreased ECM content in the marginal zone and increased cardiac contractility	[73]
**microRNA-21**	Macrophages	MicroRNA-21 inhibits ERK signaling and enhances cardiac fibroblast survival by suppressing the expression of SPRY1 in CFs	[74]
**microRNA-155**	Macrophages and fibroblasts	MicroRNA-155 inhibits cardiac fibroblast proliferation by downregulating Sos1 expression and promotes inflammation by decreasing cytokine signaling inhibitor 1 expression	[75]
**TLR2**	Macrophages	TLR2 deficiency inhibits macrophage-dependent CF activation via modulation of the TGF-β/Smad2/3 pathway	[76]

αSMA, smooth muscle α-actinin; Comp, cartilage oligomeric matrix protein; CTGF, connective tissue growth factor; HIMF, hypoxia-inducible mitogenic factor; CaMKII, Ca^2+^/calmodulin-dependent protein kinase II; CFs, cardiac fibroblasts; MO/MΦs, monocytes/macrophages; ECM, excessive extracellular matrix; SPRY1, sprouty homolog 1; Sos1, son of sevenless 1; MI, myocardial infarction.

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
