# Peer review of "Targeting Interactions between Fibroblasts and Macrophages to Treat Cardiac Fibrosis"

_cells, 2024, doi:10.3390/cells13090764_

Round 1
Reviewer 1 Report
Comments and Suggestions for Authors
This is a nice, well-written review article emphasizing the fibroblast-microphage relation with an eye for therapeutic targeting in cardiac fibrosis. The manuscript is well organized. Here are suggestions/slight concerns:
- in the composition of cells (% of cells in a heart), the number provided is not standard, meaning that there are different values reported in the literature. Thus, it would be important to emphasize that, the ranges, and specify in which mammalian heart are the numbers reported from. Ideally a range with the species would be beneficial.
- There is no mention of approaches using biomaterials to mitigate fibrosis such as myocardial matrix hydrogels.
https://pubmed.ncbi.nlm.nih.gov/35618911/
https://pubmed.ncbi.nlm.nih.gov/36041648/
https://pubmed.ncbi.nlm.nih.gov/15358036/
Author Response
- in the composition of cells (% of cells in a heart), the number provided is not standard, meaning that there are different values reported in the literature. Thus, it would be important to emphasize that, the ranges, and specify in which mammalian heart are the numbers reported from. Ideally a range with the species would be beneficial.
Thank you for your helpful feedback. We have revised the mentioned values to reflect a range, enhancing the accuracy of the data presented.
- There is no mention of approaches using biomaterials to mitigate fibrosis such as myocardial matrix hydrogels.
Thank you for your constructive comment. In accordance with your suggestion, we have introduced the section and incorporated the recommended literatures into the fourth part of the manuscript.
Reviewer 2 Report
Comments and Suggestions for Authors
The crosstalk between fibroblasts, macrophages, and ECM in cardiac tissues
is of particular interest for cardiology. In this narrative review the authors
summarized the relative evidence aiming predominantly in cardiac fibrosis treatment. It presents data from cell culture and animal experimental models and close with a presentation of potential treatments. The table and illustrations are well designed and updated.
Author Response
The crosstalk between fibroblasts, macrophages, and ECM in cardiac tissues is of particular interest for cardiology. In this narrative review the authors summarized the relative evidence aiming predominantly in cardiac fibrosis treatment. It presents data from cell culture and animal experimental models and close with a presentation of potential treatments. The table and illustrations are well designed and updated.
-Thank you for your insightful comments on our narrative review. Your appreciation of the tables and illustrations reaffirms our commitment to delivering clear and informative content.
Reviewer 3 Report
Comments and Suggestions for Authors
This review makes a very interesting point on the relationships between cardiac fibroblasts and macrophages in physiological and pathological contexts. In particular, it sheds light on the critical role of these interactions during cardiac remodeling and opens new avenues for treatment. Although the mechanisms underlying this relationship still need to be explored, the development of new tools is heralded as promising (new model of organoid MI) because they should make it possible to overcome certain limits (genetic, animal/human). Among the innovative treatments identified in this paper are the possibility of using a CAR-T strategy, the use of LNP or the use of cell therapy using the reprogramming of fibroblasts into cardiomyocytes to repopulate scar tissue and restore cardiac function.
In conclusion, very nice review, well documented with relevant illustrations.
Author Response
This review makes a very interesting point on the relationships between cardiac fibroblasts and macrophages in physiological and pathological contexts. In particular, it sheds light on the critical role of these interactions during cardiac remodeling and opens new avenues for treatment. Although the mechanisms underlying this relationship still need to be explored, the development of new tools is heralded as promising (new model of organoid MI) because they should make it possible to overcome certain limits (genetic, animal/human). Among the innovative treatments identified in this paper are the possibility of using a CAR-T strategy, the use of LNP or the use of cell therapy using the reprogramming of fibroblasts into cardiomyocytes to repopulate scar tissue and restore cardiac function.
In conclusion, very nice review, well documented with relevant illustrations.
-Thank you for your insightful comments on our narrative review. Your appreciation of this manuscript reaffirms our commitment to delivering clear and informative content.
Reviewer 4 Report
Comments and Suggestions for Authors
Dear Authors,
Please consider the following comments when revising the manuscript.
1. ROS and oxidative stress play very important roles in CF activation and inflammation. Please consider including them into the discussion.
2. Direct cell-cell interaction between fibroblast and macrophage was not discussed. e.g. cell-cell connection through cadherin. May need a new figure to discribe these interaction.
3. Some background information, such as MFGE8, was reported in lung, not heart. The fibrosis in heart is very different from that in other organs. and the cardiac fibrosis cannot be removed easily compared to other more regenerative tissue, such as liver and skin. Please add in some discussion.
4. Figure 2 did not cover all the paracrine factors discussed in the manuscript.
5. The role of IL-4 in M1/M2 needs clarification. There are conflict between the in vivo and in vitro data, please add in in-depth discussion.
6. There are available secretome studies of cardiac fibroblast and macrophages in heart, kidney etc. Please include them into the discussion. This will provide rationale of including those factors the authors chose to discussion in the manuscript.
7. The rationale to include miR21 is vague, the evidence provides don't seems to support an interaction between cardiac fibroblasts and macrophages.
Comments on the Quality of English LanguageLanguage is good. Some paragraphs will need the sentences to be re-organized to improve the flow.
Author Response
1. ROS and oxidative stress play very important roles in CF activation and inflammation. Please consider including them into the discussion.
-Thank you for your constructive comments. In response, we have expanded the Discussion section of our article to include a new subsection that highlights the significant regulatory role of reactive oxygen species in both fibroblast activation and ECM remodeling, particularly within the scope of biomaterials applications.
2. Direct cell-cell interaction between fibroblast and macrophage was not discussed. e.g. cell-cell connection through cadherin. May need a new figure to discribe these interaction.
-Thank you for your meticulous feedback. The scope of this review is specifically tailored to explore the direct and indirect interactions between fibroblasts and macrophages in relation to cardiac fibrosis, with the intent of identifying novel molecular targets for fibrotic treatment. Consequently, we have not delved into the molecular events pertaining to the intercellular junctions between these two cell types.
3. Some background information, such as MFGE8, was reported in lung, not heart. The fibrosis in heart is very different from that in other organs. and the cardiac fibrosis cannot be removed easily compared to other more regenerative tissue, such as liver and skin. Please add in some discussion.
-Thank you for your valuable suggestion. We’ve updated the section accordingly to clarify that the MFGE8 study pertains to lung tissue. Furthermore, we have included a note at the end of this paragraph to emphasize the distinction between cardiac fibrosis and fibrotic processes in other organs.
4. Figure 2 did not cover all the paracrine factors discussed in the manuscript.
-Thank you for your insightful suggestion. We have revised Figure 2 to incorporate miR-21. It’s important to note that while other molecules not depicted in Figure 2 do play a role in regulating cardiac fibrosis, they do so by indirectly modulating the interactions between fibroblasts and macrophages. For instance, IL-17A prompts chemokine production by cardiac fibroblasts (CFs), which in turn attracts neutrophils and Ly6Chigh monocytes/macrophages (MO/MΦs) to the heart. The direct molecular interactions between fibroblasts and macrophages remain less defined, which is why they are not illustrated in Figure 2.
5. The role of IL-4 in M1/M2 needs clarification. There are conflicts between the in vivo and in vitro data, please add an in-depth discussion.
-Thank you for your thorough review. We have carefully revised the section in question and have included a discussion of potential explanations for the outcomes observed in both the in vitro and in vivo experiments.
6. There are available secretome studies of cardiac fibroblast and macrophages in heart, kidney etc. Please include them into the discussion. This will provide rationale of including those factors the authors chose to discussion in the manuscript.
-Thank you for your valuable suggestion. We have incorporated references on secretome studies of cardiac fibroblasts and cardiac macrophages into the third section of our article.
7. The rationale to include miR21 is vague, the evidence provides don't seems to support an interaction between cardiac fibroblasts and macrophages.
-Thank you for your attentive review. Our discussion was indeed primarily based on the pivotal 2021 study “MicroRNA-21-Dependent Macrophage-to-Fibroblast Signaling Determines the Cardiac Response to Pressure Overload. Circulation 2021, 143, 1513-1525,” and we acknowledge that our initial description did not fully convey the importance of this research. To address this, we have enriched the relevant section with additional details. Moreover, miR-21 has now been included in Figure 2 to visually represent its central role. The new text reads: “Furthermore, a systematic assessment of ligand-receptor-mediated intercellular communication across all cell types has revealed that miR-21 is a predominant regulator of macrophage-fibroblast signaling, and it facilitates the transformation of quiescent fibroblasts into active myofibroblasts. Similarly, macrophages cultured in vitro have demonstrated equivalent capabilities in modulating fibroblast activation via miR-21”
Reviewer 5 Report
Comments and Suggestions for Authors
NA
Comments on the Quality of English Language
Cardiac fibrosis is the excessive deposition of extracellular matrix (ECM) proteins in cardiac tissue, leading to structural changes and impaired function of the heart. Several factors contribute to cardiac fibrosis, including chronic inflammation, oxidative stress, mechanical stress, and neurohormonal activation. Understanding the mechanisms underlying cardiac fibrosis and ECM remodeling is essential for developing effective therapeutic strategies to prevent or reverse fibrosis and its associated cardiovascular complications. In this review, the authors discuss emerging interactions between fibroblasts and macrophages under physiological and pathological conditions, providing insights for future studies targeting macrophages against cardiac fibrosis. Overall, this is an interesting and informative paper. However, there are still a couple of issues that need to be improved upon by the authors.
1. A diagram illustrating the origin of cardiac fibroblasts and cardiac macrophages and their differences is desirable.
2. More descriptions of M1 and M2 macrophages are needed, such as their origin, function and how to differentiate between them, etc.
3. It is recommended that a table be used to highlight the molecular mechanisms associated with the interaction between cardiac fibroblasts and cardiac microphages. This will make it easier for the reader to read and understand.
4. Authors are advised to review the manuscript again carefully to check for typos, grammatical errors, and redundant sentences for ease of reading.
Author Response
1. A diagram illustrating the origin of cardiac fibroblasts and cardiac macrophages and their differences is desirable.
-Thank you for your insightful comment, which has contributed to enhancing our manuscript. In response, we have incorporated diagrams depicting the origins of cardiac fibroblasts and cardiac macrophages into Figures 2 and 3, respectively, to provide a clearer understanding of these cellular pathways in the context of the article.
2. More descriptions of M1 and M2 macrophages are needed, such as their origin, function and how to differentiate between them, etc.
-Thank you for your constructive suggestion. We have incorporated the content into Section 2.2 of the article, where we address the nuanced heterogeneity of macrophages beyond the traditional M1/M2 macrophages classification, which is now considered overly simplistic. This discussion sets the stage for a deeper exploration of macrophage origins, classification, and their diverse functions within the cardiac environment, which is elaborated upon in a subsequent section of the article.
3. It is recommended that a table be used to highlight the molecular mechanisms associated with the interaction between cardiac fibroblasts and cardiac macrophages. This will make it easier for the reader to read and understand.
-Thank you for your meticulous review. To facilitate ease of reference, we have now consolidated the complete array of molecular mechanisms into Table 2.
4. Authors are advised to review the manuscript again carefully to check for typos, grammatical errors, and redundant sentences for ease of reading.
-Thank you for your valuable suggestion. Following your recommendation, we have conducted a thorough double-check of the manuscript for accuracy and corrected any typos and grammatical errors. Additionally, the article has been reviewed by a native English-speaking expert to ensure linguistic precision.